# Aminoglycoside-Modifying Enzymes Are Sufficient to Make *Pseudomonas aeruginosa* Clinically Resistant to Key Antibiotics

**DOI:** 10.3390/antibiotics11070884

**Published:** 2022-07-01

**Authors:** Aswin Thacharodi, Iain L. Lamont

**Affiliations:** Department of Biochemistry, University of Otago, Dunedin 9054, New Zealand; thaas007@student.otago.ac.nz

**Keywords:** aminoglycoside-modifying enzymes, aminoglycoside resistance, *Pseudomonas aeruginosa*, efflux pumps, horizontal gene transfer, tobramycin, amikacin, gentamicin

## Abstract

Aminoglycosides are widely used to treat infections of *Pseudomonas aeruginosa*. Genes encoding aminoglycoside-modifying enzymes (AMEs), acquired by horizontal gene transfer, are commonly associated with aminoglycoside resistance, but their effects have not been quantified. The aim of this research was to determine the extent to which AMEs increase the antibiotic tolerance of *P. aeruginosa*. Bioinformatics analysis identified AME-encoding genes in 48 out of 619 clinical isolates of *P. aeruginosa*, with *ant(2′)-Ia* and *aac(6′)-Ib3*, which are associated with tobramcyin and gentamicin resistance, being the most common. These genes and *aph(3′)-VIa* (amikacin resistance) were deleted from antibiotic-resistant strains. Antibiotic minimum inhibitory concentrations (MICs) were reduced by up to 64-fold, making the mutated bacteria antibiotic-sensitive in several cases. Introduction of the same genes into four antibiotic-susceptible *P. aeruginosa* strains increased the MIC by up to 128-fold, making the bacteria antibiotic-resistant in all cases. The cloned genes also increased the MIC in mutants lacking the MexXY-OprM efflux pump, which is an important contributor to aminoglycoside resistance, demonstrating that AMEs and this efflux pump act independently in determining levels of aminoglycoside tolerance. Quantification of the effects of AMEs on antibiotic susceptibility demonstrates the large effect that these enzymes have on antibiotic resistance.

## 1. Introduction

*Pseudomonas aeruginosa* is an opportunistic pathogen that causes a broad array of acute and chronic life-threatening infections with a high rate of mortality and morbidity in immunocompromised individuals [1]. Aminoglycosides (AGs) such as tobramycin, gentamicin and amikacin are key components of the antipseudomonal antibiotic regimens being used to treat a range of infections, including endocarditis, bacteremia and pulmonary infections in bronchiectasis and cystic fibrosis (CF) patients [2,3,4]. As with other species [5], a challenge in managing *P. aeruginosa* infections is the high capacity of this species to resist antibiotics through acquired and intrinsic resistance mechanisms [6].

Acquired resistance in *P. aeruginosa* is multifactorial, involving chromosomal mutations and also genes acquired by horizontal gene transfer [7,8]. Aminoglycoside susceptibility is influenced by an efflux system, MexXY/OprM, that expels AGs from the bacterial cells [9,10,11]. Expression of *mexXY* genes is controlled by a repressor protein, MexZ [12,13,14]. Mutations in *mexZ* arise frequently in clinical isolates [15,16], resulting in increased expression of the MexXY/OprM efflux system and reduced susceptibility to AGs [17,18,19,20,21,22]. Mutations in *fusA1*, which encodes elongation factor G, are also associated with reduced susceptibility to AGs [23,24,25].

Horizontal gene transfer promotes the dissemination of aminoglycoside and other resistance genes among clinical isolates of *P. aeruginosa*, an increasing cause of concern in recent years [26,27,28,29]. Acquired resistance genes encode aminoglycoside-modifying enzymes (AMEs) that inactivate AGs by catalysing modifications at OH or NH_2_ groups of the 2-deoxystreptamine sugar moieties by phosphorylation (aminoglycoside phosphoryltransferases [Aph]), acetylation (aminoglycoside acetyltransferases [Aac]) or adenylation (aminoglycoside nucleotidyltransferases/ adenylyltransferase [Ant]) [7,29]. Genes encoding these modifying enzymes have the potential to spread amongst and between species as they are often located on mobile genetic elements such as plasmids, integrons, transposons, insertion sequences, phages and integrative and conjugative elements [30,31].

Isolates of *P. aeruginosa* that are resistant to aminoglycosides often contain AMEs [28,29,32]. However, to the best of our knowledge, the extent to which AMEs increase tolerance to aminoglycosides in clinical isolates and how their effects are influenced by the MexXY-OprM efflux pump has not been quantified. The aim of this research was to quantify the contributions of three different AMEs, Ant (2″)-Ia, Aac (6′)-Ib3 and Aph (3′)-VIa, to aminoglycoside tolerance and to understand their association with MexXY-mediated resistance and other chromosomal mutations.

## 2. Results

### 2.1. Identification of Horizontally Transferred Resistance Genes

A genome dataset of 619 clinical isolates and 172 environmental isolates of *P. aeruginosa* (Appendix A), collectively representing the genetic diversity of *P. aeruginosa*, was examined for the prevalence of acquired antibiotic resistance genes using ResFinder and RGI. Forty-nine of the clinical isolates had acquired resistance genes, with genes encoding aminoglycoside-modifying enzymes (AMEs) being the most frequent (present in 48 of these isolates) and with 25 different AMEs being identified (Figure 1A) (Appendix A). AMEs were also the most common in the environmental isolates (six different AMEs, present in seven isolates). The AMEs included nucleotidyltransferases, phosphotransferases and acetyltransferases. Many of the AMEs in clinical isolates increase tolerance to at least one of the clinically important antibiotics—tobramycin, gentamicin and amikacin—that are commonly used for treating *P. aeruginosa* infections.

β-Lactamases were the second most common class of acquired resistance genes being present in 13 of the clinical isolates (Figure 1A). Genes encoding oxacillinases (*blaOXA*), which hydrolyze and impart resistance to oxacillin [32], were the most frequent (11 isolates). However, a metallo-β-lactamase VIM-2, was the only acquired β-lactamase identified in the environmental isolates. Tetracycline and quinolone resistance genes were identified in 10 and 9 clinical isolates, respectively (Figure 1A), but were not present in any of the environmental isolates. None of the analysed genomes had acquired colistin resistance genes. As expected [33,34], the strains were clustered into two groups on an unrooted tree (Figure 1B). The majority of isolates with acquired resistance genes (44 isolates) were in group I, which includes reference strain PAO1 and the well characterised strains DK2 and LESB58. All of the environmental isolates with acquired resistance genes were in group I. Group II, which includes the well-characterised strain PA14 [35], was smaller for the genomes in this study, with only 12 isolates in this group having acquired resistance genes.

Clinical and environmental isolates containing horizontally-transferred resistance genes were phylogenetically separated and were broadly dispersed across the phylogeny, indicating the acquisition of resistance genes had occurred independently on multiple occasions (Figure 1A,B). Seven of the reference panel genomes had also acquired resistance genes, encoding AMEs (seven isolates), β-lactamases (two isolates) and quinolone resistance (two isolates).

### 2.2. Organisation of Genes Encoding AMEs

The most frequently acquired AME genes encoded N-acetyltransferases (11 types, *aac* genes) followed by O-nucleotidyltransferases (8 types, *ant* genes) and O-phosphotransferases (6 types, *aph* genes). To better understand the relationship between these genes and acquired resistance genes for other antibiotics, complete genome assemblies were obtained through long-read sequencing for the eight AME-containing clinical isolates harboring β-lactamases (eight isolates) and genes conferring resistance to quinolones (six isolates) or tetracycline (one isolate). Analysis with MobileElementFinder showed that all of the resistance genes in these isolates were carried within unit or composite transposons (Appendix A) indicating that clinical isolates harboring AMEs are commonly on transposons that carry other resistance genes.

Genes encoding Aac (6′)-Ib3 or Aac (6′)-33 aminoglycoside acetyl transferases, Ant (2′)-Ia aminoglycoside nucleotidyltransferase or the Aac (6′)-Ib-cr enzyme, which acetylates fluoroquinolones as well as tobramycin and amikacin, were commonly adjacent to β-lactamase-encoding genes (Figure 2). The finding that *aac (6′)* and *ant(2″)-Ia* genes are adjacent to β-lactamase-encoding genes is consistent with earlier studies of *E.coli* [36]. However, none of the seven O-phosphotransferase-encoding *aph* genes were adjacent to genes conferring resistance to other antibiotics. In 5 of the 8 isolates, AME-encoding genes were co-located with an integrase gene (*intI1*) that encodes a site-specific recombinase that plays a key role in acquisition of resistance gene cassettes by integron system (Figure 2).

### 2.3. MICs of Clinical Isolates with Frequent AMEs

Although AMEs are associated with resistance to aminoglycosides, the magnitude of their influence on aminoglycoside MICs has received little attention. To address this issue, we quantified the effects of AMEs from three different families that act on tobramycin, gentamicin and amikacin. Ant (2″)-Ia was the most frequent nucleotidyltransferase in our collection (17 isolates) and Aac (6′)-Ib3 was the frequent acetyltransferases (8 clinical isolates). Both enzymes have previously been reported in isolates that are resistant to tobramycin and gentamicin [37,38]. We also investigated Aph (3′)-VIa, a phosphotransferase, which acts on amikacin. This gene has been reported frequently in *P. aeruginosa* [7,39] though was present in only 1 isolate in our collection.

The MICs of 14 isolates with Ant (2″)-Ia, Aac (6′)-Ib3 and Aph (3′)-VIa were measured for tobramycin, gentamicin and amikacin (Table 1). Twelve of these isolates had MICs for tobramycin and gentamicin higher than the clinical breakpoints, consistent with these genes contributing to tobramycin and gentamicin resistance. The sole isolate with Aph (3′)-VIa was resistant to amikacin, the substrate of this enzyme, although 7 of the 14 isolates lacking this gene were also amikacin-resistant, presumably due to the existence of other AMEs that modify amikacin, such as Aac (6′)-31 and Aac (6′)-33 [40], or the effect of other genetic variations such as mutations in *mexZ*, which can contribute to amikacin resistance [11].

### 2.4. Deleting AME-Encoding Genes in Clinical Isolates

Though most of the tested isolates with AMEs were aminoglycoside resistant, the MIC varied amongst isolates and the extent to which the MIC was increased by the presence of an AME was not known. To quantify the contributions of AMEs to resistance, the *ant (2′)-Ia*, *aac (6′)-Ib3* and *aph (3′)-VIa* genes were deleted from isolates with a single copy of *ant (2′)-Ia* (1257147) or single copies of both *aac (6′)-Ib3* and *aph (3′)-VIa* (1260990).

Deleting *ant (2″)-Ia* enhanced tobramycin and gentamicin sensitivity by 2- and 8-fold (Figure 3A), although the bacteria remained clinically resistant. Complementing the mutation with the cloned *ant (2″)-Ia* gene restored the tobramycin MIC to that of wild-type, and the gentamicin MIC to twofold higher than wild-type. Deletion of *ant (2″)-Ia* had no effect on the MIC for amikacin.

Deleting *acc (6′)-Ib3* had a more marked effect, increasing tobramycin and gentamicin sensitivity by 64- and 16-fold, respectively, and lowering the MICs below clinical resistance breakpoints while having no effect on amikacin resistance. Complementation with the cloned gene resulted in a wild-type MICs for gentamicin and a fourfold increase in tobramycin MICs over wild-type (Figure 3B).

In contrast, deleting *aph (3′)-VIa* from isolate 1260990 increased amikacin sensitivity by 32-fold, lowering the MIC below the clinical breakpoint while having no effect on susceptibility to tobramycin or gentamicin. Complementation with the cloned gene resulted in a twofold increase in resistance above wild-type (Figure 3C).

Engineering a double AME mutant (∆*acc (6′)-Ib3* ∆ *aph (3′)-VIa*) in isolate 1260990 increased aminoglycoside susceptibility for all three of the tested antibiotics, as expected from the effects of the single-gene deletions (Figure 3D). Collectively, these data show that AMEs are a major contributor to levels of aminoglycoside tolerance in the isolates studied here.

### 2.5. The Effects of Introduced AMEs in AME-Free P. aeruginosa

Although we were able to delete AMEs from isolates 1260990 and 1257147, many non-reference isolates of *P. aeruginosa* are refractory to engineering of mutations into the chromosome ([41] as well as unpublished observations). To more broadly investigate the effects of AMEs, the cloned *ant (2′)-Ia*, *aac (6′)-Ib3* and *aph (3′)-VIa* genes were transformed into reference strain PAO1 and three CF isolates without any AMEs. The CF isolates were chosen to have an active MexXY efflux system and sequence polymorphisms in the MexZ and FusA1 genes, which are known to increase MexXY production and aminoglycoside tolerance. The selected CF isolates had 5- to 77-fold higher *mexXY* expression than strain PAO1 [42]. The cloning vector alone did not alter the MIC for any of the antibiotics (Appendix A).

Expression of *ant (2″)-Ia* in strain PAO1 and the 3 CF isolates resulted in an increase of between 16- to 64-fold in MIC for tobramycin and gentamicin (Figure 4A). The cloned *aac (6′)-Ib3* gene increased tobramycin and gentamicin MICs by between 16- to 64-fold (Figure 4B). As expected, these cloned genes had no effect on amikacin MIC. Conversely, the cloned *aph (3′)-VIa* gene resulted in 8- to 64-fold increases in amikacin MIC while having no effect on the MIC for tobramycin or gentamicin. The findings from expression of AMEs in strain PAO1 and the three CF isolates are consistent with the effects of deletion of AMEs (Figure 3) and with the known aminoglycoside targets of these AMEs (Appendix A).

### 2.6. The Effects of AMEs in Combination with Other Resistance Mechanisms

AMEs and mutations affecting the activities of other resistance-associated mechanisms, primarily the MexXY efflux system, are expected to act independently in reducing susceptibility to aminoglycosides because of their different modes of action. To investigate this relationship, the cloned AME-encoding genes were transformed into mutants of strain PAO1 and the three AME-lacking clinical isolates from which the *mexXY* genes had been deleted. MICs were determined to assess the effects of the cloned AME genes in the absence of MexXY efflux (Table 2).

The presence of the cloned *ant (2″)-1a* and *aac (6′)-Ib3* genes increased the MIC for tobramycin and gentamicin by 4- to 64-fold in all of the *mexXY* mutants and *aph (3′)-VIa* increased amikacin MIC by 32- to 128-fold. These increases were similar to those observed in the isogenic Mex+ bacteria (Figure 3). The cloned AME genes were sufficient to render the *mexXY* mutants resistant to aminoglycosides in most cases.

The effects of the plasmid-borne AME-encoding genes were also investigated in a PAO1 *mexZ* mutant that has increased expression of the MexXY efflux pump and a PAO1 mutant carrying a mutation R680C in *fusA1* that arises frequently during chronic *P. aeruginosa* infections and is associated with aminoglycoside resistance. The cloned *ant (2″)-1a* and *aac (6′)-Ib3* genes in the *mexZ* mutant increased the MICs for gentamicin by 16- to 128-fold (Table 3). The presence of the cloned *aph (3′)-VIa* gene in this mutant caused a 128-fold increase in amikacin MICs. Similar increases in MIC occurred following transformation of the plasmids into a PAO1 *fusA1*_R680C_ mutant (Table 3).

## 3. Discussion

Resistance to aminoglycosides of *P. aeruginosa* has been a rising concern in recent years. *P. aeruginosa* can become resistant to aminoglycosides through chromosomal mutations and by acquisition of resistance genes through horizontal gene transfer [34,43]. The most frequent AME-encoding genes in our survey of over 700 genomes were *ant(2″)-I* and *acc(6′)-I*, which are commonly found in integrons and that mediate resistance to gentamicin and tobramycin [44,45,46], and *aph (3′)*-VI, which is present in transposons in isolates resistant to amikacin [47,48,49]. The same genes were the most frequent AME-encoding genes in aminoglycoside-resistant isolates of *P. aeruginosa* in other studies [40,50,51] underscoring their importance in aminoglycoside resistance in this species. Multiple AME genes were often located on the same mobile genetic elements along with genes associated with resistance to other classes of antibiotics, and some isolates contained multiple copies of the same AME gene (Figure 2; Appendix A).

The presence of AMEs is often associated with resistance. However, so far as we are aware, deleting AME-encoding genes in order to quantify the contributions of these enzymes to aminoglycoside tolerance has not been carried out previously. This is likely due to the technical challenges associated with the repetitive DNA sequences that flank many AME genes and the difficulty of genetically manipulating many clinical isolates of *P. aeruginosa*. Deleting AMEs reduced the MIC by 4- to 64-fold, indicating the contributions of these AMEs to the isolates containing them. Deleting the *aac (6)-Ib3* gene from isolate 1260990 had a bigger effect on the MICs for tobramycin and gentamicin (32- to 64-fold reduction) than deleting the *ant (2″)-1a* gene from isolate 1257147 (4- to 8-fold reduction). This difference is likely due at least in part to the presence of other AMEs in isolate 1257147, reflected in higher MICs of the 1257147 *ant (2″)-1a* mutant than the 1260990 *aac(6′)-Ib3* mutant. Indeed, isolate 1260990 became clinically sensitive following deletion of *aac(6′)-Ib3*, demonstrating the contribution of the gene to resistance, but isolate 1257147 remained clinically resistant following deletion of the *ant(2″)-1a* gene. Deleting the *aph (3′)-VIa* gene from isolate 1260990 resulted in a 32-fold reduction in the MIC for amikacin, consistent with amikacin being the substrate for the Aph (3′)-VIa enzyme and showing that the presence of *aph (3′)-VIa* is sufficient to make this isolate clinically resistant to amikacin. A double mutant, 1260990 ∆*aph (3′)-VIa* ∆*aac (6′)-Ib3*, in which two different AMEs were deleted was sensitive to all three tested aminoglycosides, consistent with the proposal that AMEs are a primary cause of clinical levels of aminoglycoside resistance in clinical settings [46].

Complementation of the deletion mutations with cloned genes restored MICs to levels the same as, or slightly higher than, wild-type. Higher MICs may reflect higher expression of the cloned genes than the initial chromosomal versions, although in several cases the cloned genes did not increase the MIC above wild-type suggesting that any difference in gene expression need not artificially raise the MIC. To mimic the effects of horizontal gene transfer, we introduced the cloned genes into isolates lacking mobile genetic elements. *Ant (2″)-Ia* and *aac6′-Ib3*, which are associated with gentamicin and tobramycin resistance [7,52] (Figure 3), conferred resistance to these antibiotics on all four isolates tested, making all of them clinically resistant while having no effect on amikacin susceptibility [50,52]. Similarly, *aph (3′)-VIa*, a frequent AME determinant in amikacin-resistant isolates of *P. aeruginosa* [7,29,40,51], made all four tested isolates resistant to this antibiotic while having no effects on tobramycin and gentamicin MICs. Collectively, these findings demonstrate the high impact of AMEs on aminoglycoside resistance.

Different isolates with the same AMEs had different MICs (Table 1) reflecting the impacts of sequence variants in other genes, in particular those listed in Table 1, on antibiotic tolerance. Nonetheless, clinical isolates with AMEs were all clinically resistant to the corresponding antibiotic, with two exceptions (isolates 1607533 and 1275655). Both of these isolates have a premature stop codon in *mexX* that likely renders the efflux pump inactive, suggesting that an AME may not be sufficient to confer clinical resistance in the absence of a functional MexXY efflux pathway. To more fully understand the interplay between AMEs and the *mexXY* efflux system the cloned AME-encoding genes were transferred into mutants lacking *mexXY* efflux pump genes or lacking the *mexZ* gene, thus having increased expression of *mexXY*. The fold increase in MIC in the *mexXY*, *mexZ* and *fusA1* mutants was similar in almost all cases to the fold increase in wild-type bacteria when the cloned genes were present, showing that the effects of AMEs are additive with those of other resistance mechanisms. In contrast to isolates 1607533 and 1275655, in most cases the presence of an AME was sufficient to make the bacteria clinically resistant to the corresponding antibiotic even in the absence of a functional MexXY efflux pump. It has been suggested that expression of *mexXY* is correlated with the presence of AMEs and that MexXY may contribute to the expulsion of modified aminoglycosides [51]. However, our findings indicate that AMEs and MexXY act independently in contributing to aminoglycoside tolerance and we have no evidence that AME effectiveness is influenced by the level of expression of *mexXY*. The interesting question of how modified aminoglycosides are expelled from *P. aeruginosa* remains to be answered.

In conclusion, our data quantify the impact of AMEs on the MICs of *P. aeruginosa* and show that AMEs can be sufficient to make aminoglycoside-susceptible isolates resistant. They also show that the effects of AMEs are independent of, and additive with, those of other resistance mechanisms.

## 4. Materials and Methods

### 4.1. Bacterial Strains and Growth Conditions

Bacterial strains and plasmids used in the study are listed in Appendix A. Bacteria were grown overnight for 18 h at 37 °C in Luria–Bertani (LB) broth [53] at 120 rpm. Prior to conjugation, *P. aeruginosa* was grown at 42 °C without aeration in LB broth supplemented with 0.4 % potassium nitrate. δ-Aminolevulinic acid (ALA) (50 µg/mL) and tetracycline (12.5 µg/mL for *E. coli*, 24–72 µg/mL for *P. aeruginosa*) were added to the media as required.

### 4.2. Minimum Inhibitory Concentration (MIC) Testing

MIC testing was conducted using the doubling dilution technique [54]. In brief, bacterial overnight cultures grown in LB broth were diluted to 10^6^ CFU/mL and 5 μL aliquots were spotted onto Muller–Hinton agar plates (BD Difco, Franklin Lakes, NZ, USA) containing amikacin (Merck, Auckland, New Zealand), tobramycin (Mylan New Zealand Ltd., Auckland, New Zealand) or gentamicin (Pfizer New Zealand Ltd., Auckland, New Zealand). Control agar plates had no antibiotic supplementation. Plates were incubated overnight at 37 °C. The lowest antibiotic concentration that inhibited growth was taken as the MIC. Bacteria were categorised into resistant and sensitive phenotypes following CLSI guidelines [55]. Isolates categorized as having intermediate resistance were treated as meeting the threshold level of resistance. Tobramycin and gentamicin had resistance breakpoints of 8 μg/mL and amikacin had a breakpoint of 32 μg/mL.

### 4.3. Genetic Manipulations

PCR amplification using appropriate primers (Appendix A) was performed using high-fidelity Q5 polymerase (New England Biolabs, Ipswich, MA, USA). PCR products were purified using the PCR clean-up and gel extraction kit (Macherey Nagel, Dueren, Germany) prior to cloning. Enzymes from New England Biolabs were used to perform restriction endonuclease digestions and DNA cloning using standard techniques [56]. The Roche (Basel, Switzerland) High Pure plasmid extraction kit was used to extract plasmid DNA from *E. coli*, and the UltraClean Microbial Kit (Qiagen, Hilden, Germany) to extract genomic DNA from *P. aeruginosa*, following the manufacturers’ instructions.

Mutants were engineered in *P. aeruginosa* using the two-step allelic exchange method [57]. DNA fragments of between 500 and 1200bp flanking the targeted deletion/ mutation sites were amplified by PCR from the genomic DNA of the strains to be mutated. The amplified flanking fragments were cloned into an allele exchange vector, pEX18Tc [58]. DNA sequencing with M13 universal primers was used to confirm that no unintended mutations were present. The donor stain *E. coli* ST18 [59] carrying the desired construct was conjugated with *P. aeruginosa* as described previously [42] and the resulting transconjugants were selected using isolate-appropriate concentrations of tetracycline (24–72 µg/mL). Mutants containing deletions were detected using PCR with deletion-spanning primers and intended point mutations were confirmed by DNA sequencing using mutation-specific screening primers.

### 4.4. Induced Expression of Aminoglycoside Modifying Enzymes

Genes encoding *ant (2″)-Ia*, *aac (6′)-Ib3* and *aph (3′)-VIa* were amplified by PCR from *P. aeruginosa* strains 1257147 and 1260990 using appropriate primers (Appendix A). The amplified genes were cloned into the integrating arabinose-inducible expression vector pSW196 [60]. The resulting plasmids were transferred to *P. aeruginosa* isolates by conjugation from *E. coli* ST18 and transconjugants were selected on LB agar supplemented with tetracycline (24–72 µg/mL). The presence of AME-encoding genes in transconjugants was confirmed by PCR amplification with gene-specific primers. Expression was induced by inclusion of arabinose (5 mg/mL) in the growth medium.

### 4.5. Genome Analysis

Genomes of 619 clinical isolates and 172 isolates from the general environment (Appendix A) were analysed in the study. The genomes were genetically diverse, representing the full breadth of the *P. aeruginosa* species, and have been described previously [34,61,62,63]. The clinical isolates were from a range of countries and from patients with a variety of diseases, including cystic fibrosis, chronic obstructive pulmonary disease, and other serious illnesses, and included 38 isolates that were known to be multidrug resistant. The environmental isolates were primarily from the general environment with a small number from a hospital environment. Library preparation and long-read nanopore sequencing of isolates harboring mobile genetic elements were performed using an Oxford Nanopore MinION (Grandomics Biosciences Co. Ltd., Beijing, China). Genome assemblies were created using SPAdes version 3.12.0 [64]. Prokka version 1.7 was used for annotation of the assembled genomes [65].

For phylogenetic analysis, ParSNP version 1.2 from the Harvest suite 1.1.2 was used to construct a core genome alignment of clinical isolates with ILPAO1 as the reference strain [66]. The alignment generated was used to build and visualise the phylogenetic tree in iTOLv6 [67] with *P. aeruginosa* PA7, a taxonomic outlier [68], as an outgroup. Acquired resistance genes were identified using ResFinder 4.1 [69]. Findings were confirmed with Resistance Gene Identifier (RGI) version 4.2.2 using CARD database 3.0.1 [70]. MobileElementFinder v1.0.3 [71,72] was used to analyse organization of acquired resistance genes in sequenced genomes.

## Figures and Tables

**Figure 1 antibiotics-11-00884-f001:**
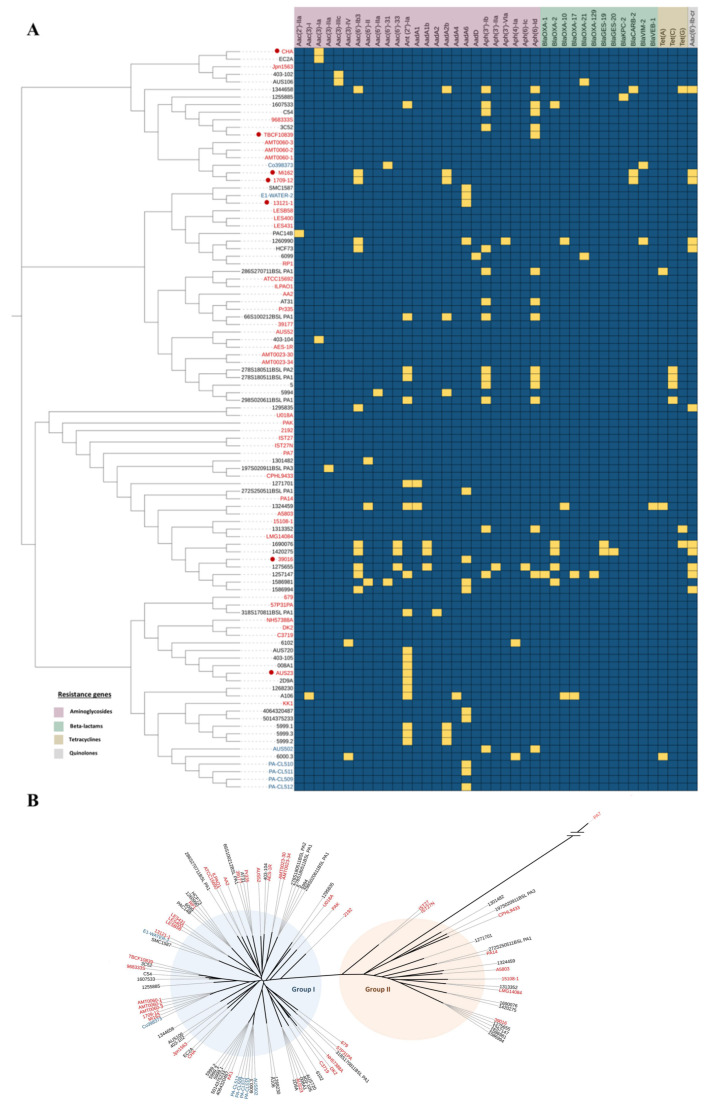
Phylogenetic analysis of isolates with acquired antibiotic resistance genes. (**A**) Antibiotic resistance gene profiling, with relationships between isolates displayed through a phylogenetic tree. The presence of a resistance gene is shown in yellow and absence in blue. (**B**) An unrooted phylogenetic tree of reference panel genomes and of genomes that contain acquired resistance genes. Forty-two reference panel genomes are shown in red with those containing horizontally transferred resistance genes indicated by red dots. Other genomes containing horizontally transferred resistance genes are shown in black (clinical) and blue (environmental). Trees were generated from whole genome sequences using ParSNP and visualised using iTOLv6.

**Figure 2 antibiotics-11-00884-f002:**
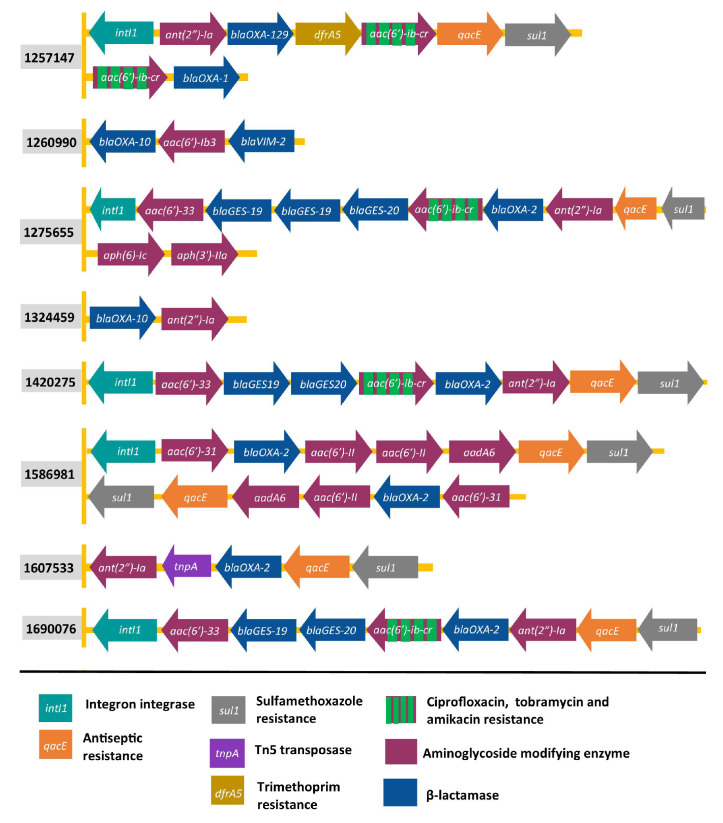
Examples of co-localisation of AME-encoding genes with genes that reduce susceptibility to other antimicrobial compounds. The presence and orientation of resistance genes in eight complete genome sequences was examined using MobileElementFinder. Each row represents a set of contiguous genes. Gene orientations are represented by arrowheads. Genes associated with transfer of mobile genetic elements (*intI1*, *tnpA*) are also shown. A complete listing of acquired resistance genes is in Appendix A.

**Figure 3 antibiotics-11-00884-f003:**
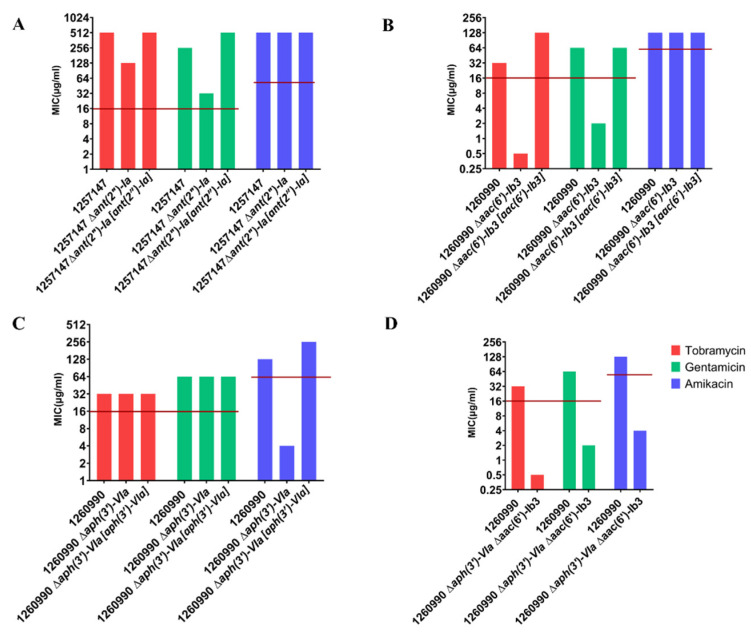
Effects of deleting AME-encoding genes on aminoglycoside resistance. (**A**) Deletion of *ant(2″)-Ia* from isolate 1257147. (**B**) Deletion of *aac (6)-Ib3* deleted from isolate 1260990. (**C**) Deletion of *aph(3′)-VIa* from isolate 1260990. (**D**) Deletion of *aac (6)-Ib3* and *aph(3′)-VIa* from isolate 1260990. MICs of wild-type isolates, deletion (∆)-containing mutants and mutants containing cloned AME-encoding genes [ ] are shown. The MIC resistance breakpoints for each antibiotic are represented by red lines.

**Figure 4 antibiotics-11-00884-f004:**
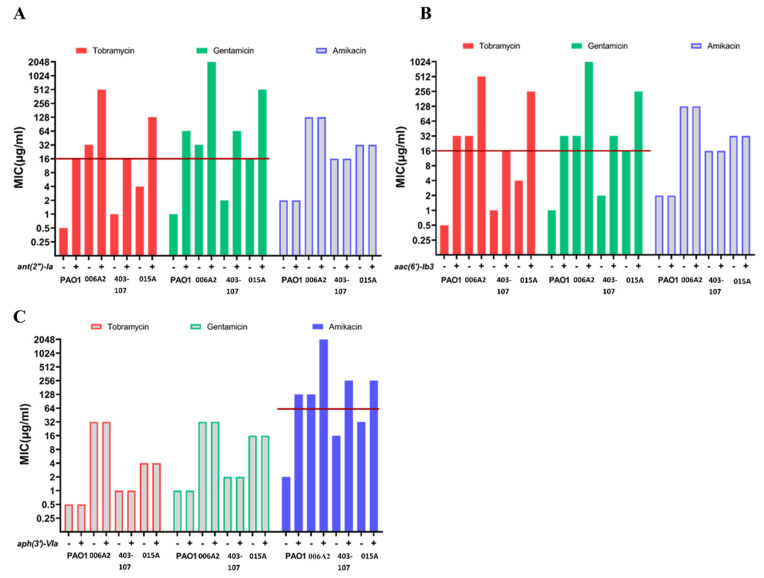
The effects of introduced AMEs on MICs. (**A**) *ant (2″)-Ia*. (**B**) *aac (6′)-Ib3.* (**C**) *aph (3′)-VIa*. The cloned genes (represented as “+”) were expressed from cloning vector pSW196 in *P. aeruginosa* strains PAO1, 006A2, 403-107 and 015A. The vector-free parental strains are represented as “−”. CLSI clinical breakpoints for MICs are indicated by red lines.

**Table 1 antibiotics-11-00884-t001:** Aminoglycoside resistance in isolates with frequently acquired AMEs.

Clinical Isolates	Source	Sequence Type ^b^	MIC ^a^			Sequence Variants	Other Acquired AMEs
			Tob	Gen	Amik		
Isolates with Ant (2″)-Ia							
403-105	CF	775	1024	512	16	*mexZ* (A38T) *armZ* (H182Q) *fusA1* (Y690C)	
008-A1	CF	775	512	128	8	*mexZ* (A38T) *armZ* (H182Q) *fusA1* (Y690C)	
1257147	Bladder	235	512	256	512	*armZ* (H182Q)	Aac (6′)-Ib Aph (6)-Id Aph (3″)-Ib
1268230	Wound	175	32	16	4	*armZ* (H182Q) *mexZ* (G195E)	
1271701	Urine	1560	16	32	8	*armZ* (H182Q)	Aph (3′)-IIb AadA1 (ANT (3″))
1324459	Burn	357	128	128	128		Aac (6′)-11 AadA1(ANT (3″))
1607533	Colon	234	2	2	4	*mexY* (E592K)	Aph (3′)- IIb Aph (3″)-Ib Aph (6)-Id
Isolates with Aac (6′)-Ib3							
1260990	Urine	395	32	64	128	*armZ* (H182Q)	AadA6 (ANT (3″)) Aph (3′) VIa
1275655	Wound	235	16	2	16	*armZ*(H182Q) *mexXY(Stop codon)*	Aac (6′)-33 AadA1b Aph (3′)-IIa Aph (6)-Ic
1295835	Sputum	646	128	128	32	*armZ* (H182Q)	
1344658	Respiratory: Endotracheal aspirate	292	256	512	256	*armZ* (H182Q) *amgS* (P139S)	AadA2b Aph (3″)-Ib Aph (6)-Id
1420275	Respiratory: Endotracheal aspirate	309	256	256	256	*mexZ* (∆6 bp)	Aac (6′)-33 AadA1b
1586994	Blood	235	256	128	256	*armZ* (H182Q) *mexZ* (∆93 bp)	AadA6 Aac (6′)-Ib-cr
1690076	Respiratory: Endotracheal aspirate	309	256	128	256	*mexZ* (∆6 bp)	Aac (6′)-33 AadA1b
Isolate with Aph (3′)-VIa							
1260990	Urine	395	32	64	128	*armZ* (H182Q)	AadA6 (ANT (3″)) Aac (6′)-Ib3

^a^ Clinical resistance breakpoints are: tobramycin, 8 µg/mL; gentamicin, 8 µg/mL; amikacin, 32 µg/mL. ^b^ MLST sequence type, determined at: https://pubmlst.org (accessed on 10 February 2022).

**Table 2 antibiotics-11-00884-t002:** The effects of AMEs on MICs in the absence of MexXY.

	Wildtype Isolate	∆*mexXY* Mutant	∆*mexXY* Mutant [*ant (2″)-Ia*]	Fold ^A^	∆*mexXY* Mutant [*aac (6′)-Ib3*]	Fold	∆*mexXY* Mutant [*aph (3′)-VIa*]	Fold
PAO1								
Tob ^B^	0.5 ^C^	0.25	8	32	16	64	0.25	0
Gen	1	0.25	16	64	4	16	0.25	0
Amik	2	0.5	0.5	0	0.5	0	64	128
006A2 ^D^								
Tob	32	1	16	16	16	16	1	0
Gen	32	1	16	16	8	8	1	0
Amik	128	4	4	0	4	0	128	32
403-107								
Tob	2	0.5	8	16	8	16	0.5	0
Gen	4	0.25	8	32	4	16	0.25	0
Amik	16	1	1	0	1	0	32	32
015A								
Tob	8	2	8	4	8	4	2	0
Gen	16	1	8	8	4	4	1	0
Amik	32	4	4	0	4	0	64	16
Fold range				4–64		4–64		16–128

^A^ MIC fold difference between the *mexXY* mutant and its AME-containing derivative. ^B^ Abbreviations: Tob, tobramycin; Gen, gentamicin; Amik, amikacin. ^C^ The clinical breakpoints are: tobramycin, 8 µg/mL; gentamicin, 8 µg/mL; amikacin, 32 µg/mL. ^D^ Isolate 006A2 has sequence variants in MexZ (T12N, Y49C) and FusA1 (Y690C). Isolate 403-107 has sequence variants in MexZ (L163P) and FusA1 (R371C). Isolate 015A has sequence variants in AmgS (R75C) and FusA1 (R680C) and has a mutation in the *mexZ* stop codon.

**Table 3 antibiotics-11-00884-t003:** Expressing aminoglycoside-modifying enzymes in the presence of additional mutations.

Antibiotics	PAO1	∆*mexZ*	*fusA1* (R680C)
Empty vector			
Tob	0.5	1	2
Gen	1	2	4
Amik	2	4	8
Expressing ***ant (2″)-IA***			
Tob	16	16	32
Gen	64	256	128
Expressing ***acc (6′)-Ib3***			
Tob	32	32	32
Gen	32	64	32
Expressing ***aph (3′)-VIa***			
Amik	128	512	256

Abbreviations: Tob, tobramycin; Gen, gentamicin; Amik, amikacin.

## Data Availability

Genome sequences of bacteria used in this study have been deposited with the National Centre for Biotechnology Information (NCBI). Accession numbers are listed in Appendix A.

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
