# Peer review of "Aminoglycoside-Modifying Enzymes Are Sufficient to Make Pseudomonas aeruginosa Clinically Resistant to Key Antibiotics"

_antibiotics, 2022, doi:10.3390/antibiotics11070884_

Round 1
Reviewer 1 Report
The article “Aminoglycoside-modifying enzymes are sufficient to make P. 2 aeruginosa clinically resistant to key antibiotics” proved by quantifying the role of aminoglycoside-modifying enzymes (AMEs) in the resistance to aminoglycosides which are widely used to treat P. aeruginosa infections. The study clearly shows that these AMEs can indeed induce high levels of antibiotic resistance in an independent way from the already described efflux pumps such as MexXY-OprM.
Regarding the manuscript I just have one question/suggestion:
- The authors correctly mentioned the different mobile elements that can do HGT. Did the authors search for prophage presence on the different isolates?
Other minor corrections:
line 15 - MICs not defined as minimum inhibitory concentration, please add it
line 76 - "..oxacillin as [32] were ..." something is missing here
Figure 1 - Can it have a better quality? The strains on A are almost unreadable
line 291 - ".. in gene expression need not inflate the MIC." please correct
Reviewer 2 Report
The manuscript is excellent as it quantified the effect of AME on antibiotic resistance, which is important for understanding the role of AME.
Reviewer 3 Report
The report by Aswin Thacarodi and Iain Lamont describes the contribution of AMEs to P. aeruginosa resistance. Through a series of elegant experiments, including deletion of AME encoding genes from clinical strains, complementation of the introduced mutations, and expression of selected AMEs in susceptible strains, they demonstrate the major (dominant) role that AMEs have in aminoglycoside resistance. This approach seems straight forward but I am also unaware of other studies showing this effect in P. aeruginosa. Additionally the authors demonstrate that the action of AMEs is independent of the major efflux pump of P. aeruginosa: MexXY. This is particularly intriguing as MexXY has been previously suggested to extrude modified aminoglycosides from the cells. The study is well planned, executed and documented and I only have a few minor remarks, which could be addressed before publication.
Minor remarks:
11. Line 64 – it might be necessary to change the order of tables in supplement as the numbering starts from Table S3.
22. Fig1 is too small in the printed version Please consider using landscape orientation for heatmap A and please explain the tree on panel A.
33. Line 107 – Have the long reads Nanopore reads been used to refine the genome assemblies? Could you provide the accession numbers to the genomes of strains used in this study (in Table S1 instead of biosamples?).
44. Table 1 – please explain “Sequence type” in footnote. Also please list the mutations (aa changes) in “Genetic variations” column.
55. Table S1 For reproducibility it might be a good idea to include accession numbers of genes that have been removed or (re) introduced or engineered in particular strains.
66. Typos: Line 393 “outliner”->outlier: Table S1 - mexZ genes -> mexZ gene
Reviewer 4 Report
The manuscript exhibited the three different gene encoding aminoglycoside-modifying enzymes (AMEs) (Ant (2")-Ia, Aac (6')-Ib3, and Aph o aminoglycoside tolerance and to understand their association with MexXY mediated resistance and other chromosomal mutations. In order to improve the quality of the manuscript, I have several comments:
1. Introduction: Some related research should be added to the Introduction. The investigation of aminoglycoside resistance from other bacteria should be provided in the introduction.
2. Figure 1 should be improved the resolution and the legend should be clear.
3. The discussion section has been corrected.
